# Effects of Bile Acids on Growth Performance and Lipid Metabolism during Chronic Heat Stress in Broiler Chickens

**DOI:** 10.3390/ani11030630

**Published:** 2021-02-27

**Authors:** Chang Yin, Shanlong Tang, Lei Liu, Aizhi Cao, Jingjing Xie, Hongfu Zhang

**Affiliations:** 1The State Key Laboratory of Animal Nutrition, Institute of Animal Sciences, Chinese Academy of Agricultural Sciences, Beijing 100193, China; 82101186178@caas.cn (C.Y.); long18763897938@163.com (S.T.); swina2010@163.com (L.L.); zhanghongfu@caas.cn (H.Z.); 2Shandong Longchang Animal Health Care Co., Ltd., Jinan 251100, China; longchang5188@126.com

**Keywords:** heat stress, bile acids, liver, lipid, broiler

## Abstract

**Simple Summary:**

The negative impacts of heat stress (HS) on growth performance and lipid metabolism have been reported, but there are still no effective nutritional strategies to alleviate heat stress. Bile acids are new for their antioxidative properties and regulatory effect on lipid metabolism. This study was carried out to evaluate the growth performance and lipid metabolism in chickens under heat stress when fed with bile acid supplements in their diet. The results showed that mild heat stress (32 °C) induced hepatic lipogenic gene (hepatic *SREBP-1c*) expressions and lipid deposition, without obvious tissue damage in broilers. Dietary supplementation of bile acid could decrease hepatic lipid deposition without affecting endogenous bile acid biosynthesis. Therefore, bile acid supplements can benefit broiler chickens during high ambient temperatures, serving as a new nutritional strategy against heat stress.

**Abstract:**

This study aimed to investigate whether dietary bile acid (BA) supplements can improve growth performance and lipid metabolism in heat-stressed broiler chickens. A total of 288 Arbor Acres broilers were blocked by BW and then randomly allocated into 4 treatments at 21 days of age. Birds reared under 32 °C had a higher cloacal temperature (*p* = 0.01), faster respiratory rate (*p* < 0.001), and a greatly reduced average daily feed intake (ADFI, *p* = 0.016), average daily gain (ADG, *p* = 0.006), final body weight (FBW, *p* = 0.008), and feed conversion rate (FCR, *p* = 0.004). In heat stress (HS) birds, the breast muscle rate (*p* = 0.006) and pH 24 h postmortem (*p* = 0.065) were lower, and the shear force was higher (*p* = 0.027). Dietary BA supplements tended to increase the breast muscle rate (*p* = 0.075) without affecting the growth performance and serum lipids (*p* > 0.05). Serum total bile acid (TBA) was roughly duplicated after BA supplements (*p* = 0.001). In the liver, total cholesterol was lower (*p* = 0.046), and triglycerides were higher (*p* = 0.04) in the HS birds, whereas the expression of *SREBP-1c* showed an increasing trend (*p* = 0.06). In contrast, dietary BA decreased triglycerides and the expressions of hepatic *SREBP-1c* and *FAS* in the liver (*p* < 0.05). In summary, mild HS causes hepatic lipid accumulation without obvious tissue damages, whereas BA has positive effects on relieving abnormal lipid metabolism, indicating that BA as a nutritional strategy has a certain potential in alleviating HS.

## 1. Introduction

High temperature is the major environmental condition posing severe threats to livestock in the summer, and billions of dollars in losses are due to heat stress (HS) annually [1]. Modern broilers are bred for their fast growth rate and superior feed conversion ratio (FCR), making them more susceptible to high temperatures during the development of skeletal muscle [2]. Heat stress negatively impacts feed intake, feed conversion, and carcass quality, leading to reduced growth performance and even growth stagnation [3]. During heat exposure, a large amount of blood flows to the body surface to maximize radiant heat dissipation [4], causing ischemia in other organs and tissues. Ischemia challenges the antioxidant system in the body. Free radicals, which cannot be eliminated in time or are beyond the ability to be eliminated, cause oxidative stress, lipid peroxidation, and mitochondrial damage [5,6]. Baumgard and Rhoads [3] indicated that lipolytic capacity is reduced in heat-stressed animals; in contrast, the capacity to uptake and store intestinal and hepatic-derived triglycerides is increased. Excessive triglycerides (TGs) were unable to be transported in chronically heat-stressed broilers because of the limited apolipoprotein B (ApoB) [7]. Consequently, heat stress can cause metabolic disorders with excessive fat deposition [8]. In chickens, the liver produces 95% of lipids [9], and abnormal hepatic fat deposition during heat stress can deteriorate in chickens [8]. Therefore, besides antioxidation, the improvement of liver function in fat metabolism should be addressed to alleviate the hazards of heat stress in chickens. Alleviating the hazards of heat stress in broiler chickens with a nutritional approach has always been a focus of the poultry industry. Although some anti-heat-stress additives are used in actual production, such as supplementing electrolytes and vitamins, the effects of improving growth performance and lipid metabolism are still not ideal [10,11]. In addition, increasing energy concentration will improve growth performance; however, increasing fat accumulation is also a major problem [12].

Bile acids (BAs) as cholesterol derivatives are essential for fat digestion and metabolism. Primary BA, cholic acid (CA), and chenodeoxycholic acid (CDCA) are synthesized in hepatocytes, conjugated with taurine or glycine, and stored in the gallbladder [13]. After eating, bile salts are secreted into the intestine and facilitate the digestion and absorption of lipid- and fat-soluble substances [14]. Around 95% of BAs reabsorbed in the terminal ileum are recycled to the liver via the enterohepatic circulation [15]. Bile acids are combined with polar phospholipid molecules in the intestinal lumen through a non-receptor-mediated mechanism, and dietary lipids are incorporated into the mixed micellar solution in the intestinal lumen [16]. Some BAs, such as CA and CDCA, can regulate glucose and lipid metabolism [17,18]. Previous study has indicated that CDCA and CA lower TG levels by targeting sterol regulatory element-binding protein *1c* (*SREBP-1c*) gene expression [18]. *SREBP-1c* regulates the fatty acid synthase (*FAS*) expression of lipogenic genes. *SREBP-1c* overexpression leads to increased levels of TG in the liver [19]. In addition, BA, ursodeoxycholic acid (UDCA), and taurine-conjugated UDCA (TUDCA) are known as chemical chaperons against endoplasmic reticulum stress and are also known to have antioxidative stress properties [20,21]. A previous study indicated that adding 200 mg/kg TUDCA to the diet of heat-stressed broilers can significantly reduce the malondialdehyde content in the serum and increase the total antioxidant capacity of the liver [22]. Liu et al. [23] also found CA can bind to the Keap1 protein through hydrogen bonding and π-π accumulation to upregulate nuclear-erythroid 2-related factor 2 (Nrf2) signal transduction and affect downstream protein expression, thereby alleviating oxidative stress. 

Based upon their regulatory effect on metabolism and antioxidative properties, it is hypothesized that BA supplements could alleviate the negative effects of heat stress. Therefore, this study was carried out to evaluate the growth performance and lipid metabolism in chickens under heat stress when fed with BA supplements in their diet.

## 2. Materials and Methods

### 2.1. Animals, Diet, and Experimental Design

Experimental procedures related to the use of live roosters were approved by the Animal Care and Welfare Committee of the Institute of Animal Sciences, Chinese Academy of Agricultural Sciences (IASCAAS). The Code of Ethical Inspection was IAS 2019-78.

The experiment adopted a 2 × 2 factorial split-plot design, with temperature (23 or 32 °C) assigned to the main plots and BA supplements (0 or 200 mg/kg of feed) assigned to the split-plots. The experiment was replicated in three blocks. The BA compound was of porcine origin, provided by Shandong Longchang Animal Health Care Co., Ltd. BA composition was analyzed by LC-MS/MS [15]: taurohyocholic acid (THCA, 34.02%), UDCA (25.58%), CDCA (16.09%), hyodeoxycholic acid (HDCA, 11.23%), and glycoursodeoxycholic acid (GUDCA, 9.60%). The verification of BAs from the compound was performed using LC-MS/MS following the method established in the early stage of the laboratory. Day-old Arbor Acres male broiler chicks were reared in cages for 16 d. On D16, all birds were weighed, and two hundred eighty-eight broilers were selected and divided into 3 blocks according to their body weight. Each block of birds was randomly allocated into two environmental control chambers with 4 floor pens (12 birds/pen) in each chamber. After 5 d adaptation in the chamber, the temperature in half of the chambers was increased to 32 °C in 2 h from 8:00 am on D21; the rest were maintained at 23 °C until D42. All birds were free to feed and water. Environmental conditions except temperature were according to the AA broiler management guide and kept the same among all chambers. The experimental diets (Appendix A) were formulated to meet or exceed the nutritional requirements for broiler chickens recommended by the NRC (1994). During the trial period, feed intake and health status were recorded daily. At the end of the trial, respiratory rate (2 birds/pen) and cloaca temperature (2 birds/pen) were measured as described [24].

On D42, all birds were fasted overnight, and body weight was measured. Blood samples were withdrawn from the wing of the bird to separate serum. The serum was stored at −20 °C until use. Before tissue sample collection, all birds were refed for 2 h, and two chickens from each pen were sacrificed by overdose anesthetic injection. The liver was removed, and samples were either snap-frozen in the liquid nitrogen or fixed with 10% formaldehyde. Breast muscle and abdominal fat were separated and weighed. Breast muscle rate and abdominal fat rate were calculated and normalized by live body weight.

### 2.2. Meat Quality Analysis

For meat quality assessment, the pH of the breast muscle samples was measured at 45 min and 24 h postmortem using a pH meter (HI 8242C, Beijing Hanna Instruments Science and Technology, Beijing, China) at three different locations. Approximately 30 g of regular-shaped breast muscle was cut off at the same position for each breast muscle sample, weighed, and recorded as the initial weight. The breast muscle was hung in a plastic seal bag at 4 °C. After 24 h, the surface water of the muscle was absorbed by filter paper, and the weight was taken and recorded as the final weight. The drip loss rate was calculated using the following equation:
Drip loss rate (%) = (initial weight – final weight)/initial weight × 100%.


To determine shear force, the breast muscle samples were placed in plastic bags and cooked in an 85 °C water bath until the internal temperature reached 77 °C. Each sample was cut into two strips (1 × 1 × 3 cm) in the myofiber direction. Each strip was measured two times using a digital meat tenderness meter (Model C1LM3, Northeast Agricultural University, Harbin, China), according to the method of Gao et al. [25]. Each sample was measured 3 times, and the average shear force value was expressed in kilograms (kg).

The breast muscle samples were freeze-dried, and the intramuscular fat (IMF) content was determined by the Soxhlet extraction method, as described by Tang et al. [26].

### 2.3. Biomarkers for Oxidative Stress and Tissue Damages

Commercial kits (Nanjing Jian Cheng Bioengineering Institute, Nanjing, China) were used to quantify malondialdehyde (MDA) and the activities of superoxide dismutase (SOD) and glutathione peroxidase (GSH-Px) in the plasma and liver tissue. The content of MDA was determined by the thiobarbituric acid method. The activity of SOD was determined by enzymatic glutathione depletion, and GSH-Px activity was measured using the xanthine oxidase method [27]. To quantify MDA, SOD, and GSH-Px in the liver, approximately 100 mg of liver tissue was homogenized in 900 mL of physiological saline, and the supernatant was harvested after centrifugation at 2000 rpm for 10 min at 4 °C. 

### 2.4. Serum and Liver Lipids

Serum TG, total cholesterol (TC), low-density lipoprotein cholesterol (LDL-C), and high-density lipoprotein cholesterol (HDL-C) were analyzed using an automatic TBA-120FR biochemical analyzer (Toshiba Corporation, Tokyo, Japan). Serum total bile acid (TBA) and free fatty acid (FFA) were accessed using commercial kits provided by Nanjing Jian Cheng Bioengineering Institute (Nanjing, China), according to the manufacturer’s instructions. Ten percent of liver homogenate was made to quantify TG and TC in the liver.

### 2.5. Oil Red O Staining of Liver Tissues

For Oil Red O staining, frozen sections were prepared (8 µm) from liver tissue samples and fixed in 50% ethanol. Subsequently, sections were stained with Oil Red O solution (Beijing Solarbio Science and Technology Co., Ltd., Beijing, China) for 8 min and differentiated with 50% ethanol, rinsed with tap water, and counterstained with hematoxylin. Following a final rinse in tap water, the sections were mounted with glycerin-gelatin-coated slides. The sections were photographed using a light microscope linked to a digital CCD camera (Model DM300, Leica, Wetzlar, Germany).

### 2.6. RNA Isolation and qPCR to Quantify Genes Related to Lipid Metabolism

Total RNA was extracted from the liver using a total RNA extraction kit (Genebetter, R013-50, Beijing, China). cDNA synthesis was performed using the PrimeScript™ RT Reagent Kit with gDNA Eraser (Takara RR047A, Dalian, China) and was subjected to qPCR amplification using SYBR Green Master Mix (Takara RR420A, Dalian, China), as described by Fang et al. [15]. Primers for critical genes related to lipid metabolism were listed as follows (Appendix A). The comparative CT method (2^−∆∆Ct^) was referred to in order to calculate the mRNA expression levels, using GAPDH as a housekeeping gene.

### 2.7. Statistical Analysis

The fitting model platform of the JMP10.0 software was used for the statistical analysis of split-plot experiment design, and the restricted maximum likelihood (REML) method was used. Data of individuals from the same subplot was considered as subsamples and averaged. In the model, the effects of temperatures (the main plot) and diets (the subplot) were fixed effects, and block and block × temperature were considered random effects. *p* < 0.05 was considered statistically significant.

## 3. Results

### 3.1. Growth and Behavioral Performance

As expected, birds reared under 32 °C displayed a heat stress response, having a higher core temperature (*p* = 0.01, Figure 1B) and faster respiratory rate (*p* < 0.001, Figure 1A). The heat-stressed birds had a lower final BW (*p* = 0.008, Figure 1C) on D42 and ADG (*p* = 0.006, Figure 1D). The average daily feed intake (ADFI) of birds reared under 32 °C was about 25% lower than those in the thermoneutral environment (*p* = 0.016, Figure 1E). The feed conversion ratio (FCR) was greatly decreased from 0.56 to 0.48 (*p* = 0.004, Figure 1F). However, BA supplements did not affect the FBW, ADG, ADFI, and FCR (*p* > 0.05). Neither cloacal temperature nor respiratory rate were affected by BA supplements (*p* > 0.05).

### 3.2. Carcass and Meat Quality

No significant difference was found in the abdominal fat rate. In birds reared at 32 °C, the breast muscle rate was about 3% lower than those kept at 23 °C (*p* = 0.006, Figure 2B), and the shear force of the breast muscle showed a slight increase with the rise of temperature (*p* = 0.027, Figure 2G). Supplementing BAs in the diet tended to increase the breast muscle rate (*p* = 0.075, Figure 2B). Regarding meat quality, exposure to 32 °C led to a decrease in the pH of the breast muscle 24 h postmortem (*p* = 0.065), but it did not influence the pH at 45 min postmortem (*p* = 0.599). BA supplements did not change pH (*p* > 0.05), but they tended to increase drip loss in the breast muscle (*p* = 0.073, Figure 2D).

### 3.3. Serum and Liver Oxidative Stress and Damages

Biomarkers for tissue oxidative damages, MDA, were not significantly changed in the serum or the liver (*p* > 0.05) by the increased environmental temperature or BA supplements (Figure 3A,D). No differences were observed in the serum and liver SOD and GSH-Px either (*p* > 0.05, Figure 3).

### 3.4. Serum, Liver Lipids, and Total Bile Acids

Compared with birds kept in the thermoneutral environment, no difference was observed in serum lipids, including TC, HDL-C, LDL-C, TG, and FFA in birds reared at 32 °C (*p* > 0.05, Figure 4). Birds fed with BA tended to have a higher HDL-C (*p* = 0.057, Figure 4B). Serum TBA was increased in birds reared under 32 °C (*p* = 0.032), and BA supplements greatly increased serum TBA in broilers (*p* = 0.001, Figure 4F).

In the liver, TC (*p* = 0.046, Figure 5A) was lower, and TG (*p* = 0.031, Figure 5B) was greater in birds kept at 32 °C than those at 23 °C. Compared with birds fed a basal diet, TC (*p* = 0.049) and TG (*p* = 0.032) were lower in birds fed a BA-supplemented diet. Oil Red O staining also showed more staining in the liver tissue of birds under 32 °C and less staining in birds fed with BA (Figure 5C–F).

### 3.5. Expression of Gene Associated with Lipid and Bile Acid Synthesis in the Liver

A significant temperature × diet effect was noted in the expression of FAS (*p* = 0.037, Figure 6B) in the liver. The expression of FAS was comparable between birds fed with the basal diet and the BA-supplemented diet at 23 °C, but it was significantly lower in birds that received the BA diet than those that received the basal diet at 32 °C (*p* = 0.011). Being reared at 32 °C, broiler chickens tended to have greater SREBP-1c (*p* = 0.06, Figure 6A), ME (*p* = 0.1, Appendix A), and ApoB (*p* = 0.105, Appendix A). Compared with their counterparts fed with the basal diet, birds fed with the BA diet were lower in SREBP-1c (*p* = 0.028) and tended to have lower ME (*p* = 0.064) expression in the liver. 

Regarding genes associated with BA synthesis (Figure 6C and Appendix A), birds kept at 32 °C were greater in hepatic FXR expression. The expression of CYP27A1 tended to be lower in birds fed with the BA diet than those fed with the basal diet. There was no difference in the gene expression of HMGCR (Appendix A), CYP7A1, and CYP8B1.

## 4. Discussion

After brooding, an environmental temperature of 32 °C can induce canonical heat stress responses in broiler chickens, increasing body temperature and accelerating respiration. Persistent exposure to high temperature decreases feed intake and growth performance. It is an adaptive response of most animal species to reduce metabolic heat production and stabilize the balance of energy production and heat loss during heat exposure [3]. Heat production will increase after eating and during the process of nutrient metabolism, which is extremely disadvantageous in high-temperature environments. Therefore, a decreased feed intake is the most intuitive impact of HS on broilers. Consequently, the growth performance of broilers cannot be guaranteed, where final BW is decreased by 25% under HS. Under thermoneutral circumstances, decreased feed intake promotes fat mobilization to maintain the energy supply. However, lipid deposition was favored in heat-stressed pigs [28] and dairy cattle [3]. To be consistent with previous findings, lipid accumulation in the liver of heat-stressed broilers was found in the present study (demonstrated by Oil Red O staining, Figure 5C–F). The increased lipid deposition was most likely due to enhanced fatty acid biosynthesis, as hepatic *SREBP-1c* and *ApoB* are upregulated in the heat-stressed birds. 

High ambient temperature impairs carcass quality, mainly via the breast muscle rate of declining and excessive fat deposition [29]. Zhang et al. [30] indicated that heat stress significantly decreases the breast muscle rate of broilers, even in diurnal cyclic high temperature and constant high temperature (34°C). Our findings noted a decrease in the breast muscle rate in broilers exposed to 32 °C, and the pH of breast muscle 24 h postmortem was also decreased. Similarly, Lu et al. [6] found broilers exposed to 32 °C for 14 d had a reduced broiler breast meat rate and pH value. Ischemia in the breast muscle during heat stress might result in accelerated meat glycolysis and increased lactate production, reducing the pH [31]. However, these heat-stressed birds did not develop obvious tissue damages, as there were no changes in serum and liver MDA and antioxidant enzyme activities. It seems that an environmental temperature of 32 °C is not sufficient to induce cellular damages. Miao et al. [32] assessed the temperature-dependent effects on the oxidative stress biomarkers and signaling in broilers. They found the levels of MDA, SOD, and GSH-PX were not altered at 29, 32, and 35°C for 6 h when Nrf2 was activated when the temperature reached 32 °C. This suggests that MDA and antioxidant enzyme activities are not sensitive as behaviors and growth performance in response to heat response.

The micellization process of BA increases intraluminal lipids in the surface area and improves the accessibility of intestinal lipase and the efficiency of fat hydrolysis. Therefore, BAs were expected to improve growth performance by facilitating fat digestion. Two previous studies showed that dietary supplements with 60a–80 mg/kg BA compound for 42 d improved the performance of broilers by elevating the activities of lipoprotein lipase and duodenum lipase but the decreased abdominal fat rate [14,33]. In the current study, however, there were no differences in the growth performance, abdominal fat rate, and serum lipids (except HDL-C) between birds fed with basal and BA-supplemented diets under a thermoneutral environment. To be consistent, genes related to lipid metabolism were not affected by BA supplements either. It was likely a 21 d dietary treatment was not sufficient for BA to improve the growth performance of heat-stressed birds because increased growth performance was only observed until 42 d [14,33]. 

Both the biochemical and histological analyses of the present study demonstrated liver fat was reduced in those BA-supplemented birds. Hepatic *FAS*, *SREBP-1c*, and *ME* encode molecules that promote fat accumulation [34], which were downregulated as well. A previous study found that 0.5% purified CDCA decreased the expression levels of *ACC-1c* and *FAS*, inhibited feed intake, and reduced body weight in broiler chickens [35]. Ge et al. [33] found BA supplements inhibited the expression of hepatic *SREBF1*, *FAS*, and *ACC* but upregulated both hepatic *CPT1* and *PPARα*. It seems that BAs may promote fat catabolism and inhibit fat synthesis by regulating lipid-metabolism-related gene expression. This lipid-lowering effect of BA supplementation is especially important in heat-stressed birds accumulating fat in the liver (Figure 5).

Naturally, BA can be synthesized from the precursor, cholesterol, in the hepatocyte via the classic and alternative routes [13]. The classic biosynthetic pathway of BA is initiated by CYP7A1 [36]. CYP8B1 is required for the synthesis of CA. The alternative pathway is initiated by CYP27A1, which is expressed in the liver as well as macrophages and most other tissues, may play important roles in total BA synthesis [37]. Prior studies have shown that diverse molecules like BA, growth factors, inflammatory cytokines, steroid hormones, and insulin can inhibit CYP7A1 transcription through the 5′-upstream region of the promoter [38,39,40,41]. Although serum TBA is almost doubled, our data have shown dietary addition of the BA compound did not affect genes coding critical enzymes involved in cholesterol and BA synthesis. This finding suggests the oral intake of 200 mg/kg BAs has few impacts on endogenous BA biosynthesis. 

High-intensity HS probably predisposes broilers to oxidative stress and endoplasmic reticulum stress [42]. However, some BAs have been demonstrated to an antioxidative effect. Xie et al. [43] showed that treatment with 200 μmol/L TUDCA could be used as a molecular chaperone to enhance protein folding and protect liver Huh7 cells against TG-induced morphologic changes of endoplasmic reticulum stress and apoptosis in vitro. Perez and Briz [44] indicated that UDCA enhances the ability to resist oxidative damage by increasing the level of glutathione in the liver and promotes immune regulation, bile secretion of liver cells, and bile duct epithelial cells. In the present study, heat exposure at 32 °C did not cause severe oxidative damages, therefore no further alleviating effect on oxidative stress was found in birds fed with BAs. In addition, since the hydrophobicity of BA determines their toxicity and protective effect [45], the BA profile of a BA compound can greatly affect its function. 

## 5. Conclusions

Dietary supplements of BA of porcine origin can significantly decrease hepatic lipid deposition without affecting the endogenous BA biosynthesis. Therefore, BA supplements can benefit broiler chickens during high ambient temperatures, serving as a new nutritional strategy against heat stress. 

## Figures and Tables

**Figure 1 animals-11-00630-f001:**
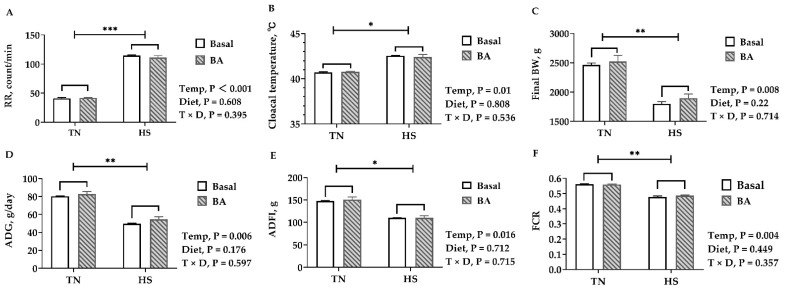
Growth and behavioral performance of broilers fed with 0 or 200 mg/kg bile acids under thermoneutral (TN) and heat-stressed (HS) conditions. (**A**) RR, respiratory rate; (**B**) cloacal temperature on D42; (**C**) final body weight; (**D**) ADG, average daily gain; (**E**) ADFI, average daily feed intake; (**F**) FCR, feed conversion rate. Data are presented as the mean ± SEM (n = 6 for each mean). * *p* < 0.05, ** *p* < 0.01, and *** *p* < 0.001.

**Figure 2 animals-11-00630-f002:**
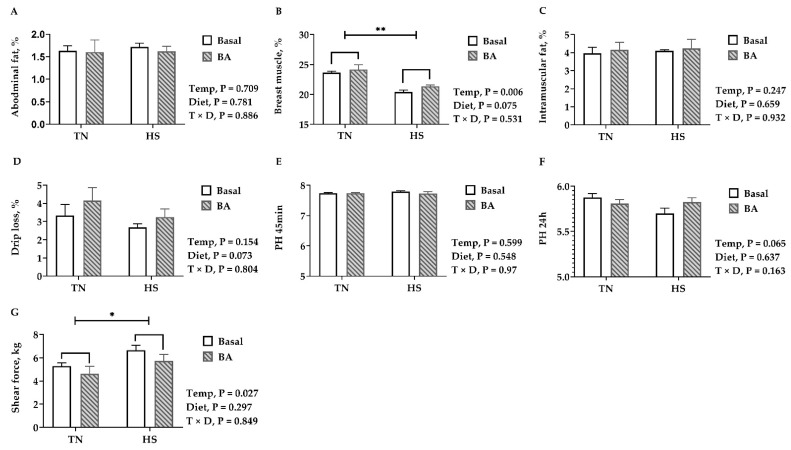
Carcass and meat quality of broilers fed with 0 or 200 mg/kg bile acids under thermoneutral (TN) and heat-stressed (HS) conditions. (**A**) Abdominal fat rate; (**B**) breast muscle rate; (**C**) intramuscular fat (freeze-dried samples); (**D**) drip Loss; (**E**) pH at 45 min postmortem; (**F**) pH 24 h postmortem; (**G**) shear force. Data are presented as the mean ± SEM (n = 6 for each mean). Different letters indicate significant differences * *p* < 0.05 and ** *p* < 0.01.

**Figure 3 animals-11-00630-f003:**
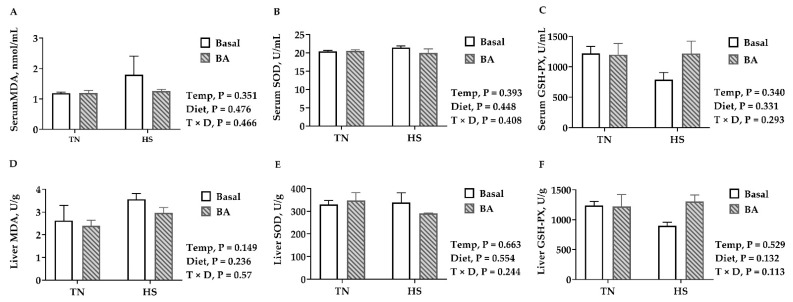
Oxidative damages parameters and antioxidative enzymes in the serum and liver of broilers fed with 0 or 200 mg/kg bile acids under thermoneutral (TN) and heat-stressed (HS) conditions. (**A**,**D**) Malondialdehyde (MDA); (**B**,**E**) superoxide dismutase (SOD); (**C**,**F**) glutathione peroxidase (GSH-Px). Data are the means ± SEM (n = 6 for each group).

**Figure 4 animals-11-00630-f004:**
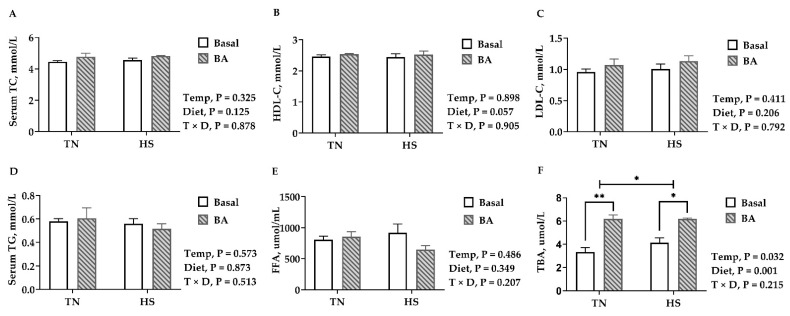
Lipid profiles in the serum of broilers fed with 0 or 200 mg/kg bile acids under thermoneutral (TN) and heat-stressed (HS) conditions. (**A**) TC, total cholesterol; (**B**) HDL-C, high-density lipoprotein cholesterol; (**C**) LDL-C, low-density lipoprotein cholesterol; (**D**) TG, triglycerides; (**E**) FFA, free fatty acid; (**F**) TBA, total bile acid. Data are the means ± SEM (n  =  6 for each group). * *p* < 0.05 and ** *p* < 0.01.

**Figure 5 animals-11-00630-f005:**
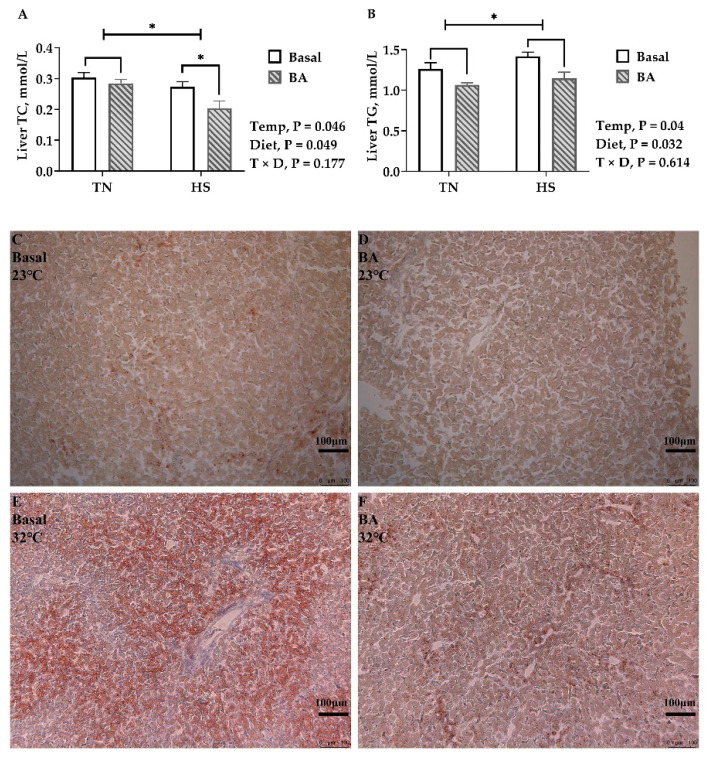
Lipid deposition in the liver of broilers fed with 0 or 200 mg/kg bile acids under thermoneutral (TN) and heat-stressed (HS) conditions. (**A**) TC, total cholesterol; (**B**) TG, triglycerides; (**C**–**F**) Representative images of Oil Red O staining (100×). Data are the means ± SEM (n = 6 for each group). * *p* < 0.05.

**Figure 6 animals-11-00630-f006:**
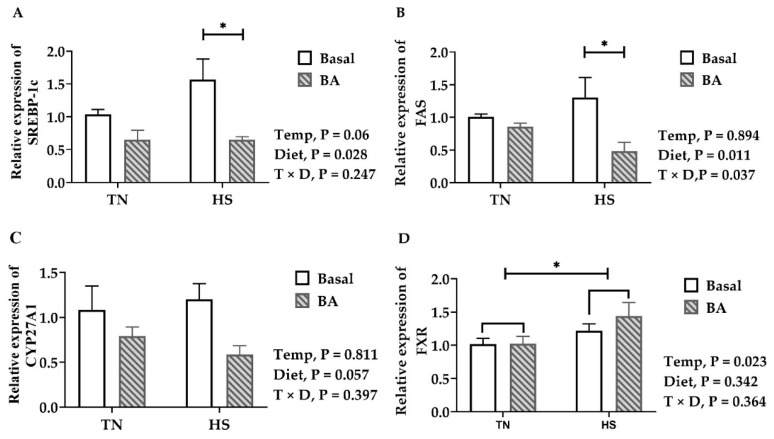
Gene expressions in the liver of broilers fed with 0 or 200 mg/kg bile acids under thermoneutral (TN) and heat-stressed (HS) conditions. (**A**) SREBP-1c, sterol regulatory element-binding protein *1c*; (**B**) FAS, fatty acid synthase; (**C**) CYP27A1, sterol 27-hydroxylase; (**D**) FXR, farsenoid x receptor. Data are the means ± SEM (n = 6 for each group). * *p* < 0.05.

## Data Availability

The data presented in this study are available on reasonable requestfrom the corresponding author.

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
