# Peer review of "Effects of Bile Acids on Growth Performance and Lipid Metabolism during Chronic Heat Stress in Broiler Chickens"

_animals, 2021, doi:10.3390/ani11030630_

Round 1
Reviewer 1 Report
The article deals with the important problem of the effect of heat stress on bird growth performance and its reduction by using bille acids in the diet.
The following inaccuracies have been observed:
24 line; 26 line; 168 line; 172 line: abbreviations (BW, ADFI, ADG, FCR, HS) are misused because they are not defined when used for the first time.
Author Response
- The article deals with the important problem of the effect of heat stress on bird growth performance and its reduction by using bille acids in the diet.
The following inaccuracies have been observed:
24 line; 26 line; 168 line; 172 line: abbreviations (BW, ADFI, ADG, FCR, HS) are misused because they are not defined when used for the first time.
Thank you. All abbreviations have been defined at the time. Please see the re-submitted manuscript in red.
Reviewer 2 Report
The topic is quite interesting. However, The author should not used abbreviations in abstract which make the readers confused and reactant to the articles.
Author Response
Thank you. All abbreviations have been defined at the time. Please see the re-submitted manuscript in red.
Reviewer 3 Report
Effects of bile acids on growth performance and lipid metabolism during chronic heat stress in broiler chickens
- It is necessary to add quantifiable data on the effect of bile acids on lipid metabolism in the summary. The abstract is written in a very general way.
- In the summary indicate what the abbreviations are and then use them. AA, HS, TBA, ADFI, ADG, BW, and FCR???.
- Add in the introduction references to the antioxidant capacity and antioxidant effect of bile acids.
- Several studies have determined the effect of the inclusion of bile acids in broiler diets on lipid metabolism and production parameters. Therefore, it is necessary to indicate some of these studies in the introduction and to specify what is the contribution of this work, with respect to the antecedents that already exist in the literature.
- In methods, more background on BA should be provided: origin (porcine? chickens?), indicate procedure of obtaining (saponification, decolorization, acidification, purification, and desiccation?). What technique was used to identify the bile acids?.
- Indicate how respiratory rate and cloacal temperature were measured.
- Add references for the following methods: thiobarbituric acid method, xanthine oxidase method.
- Line 140: change 10% by ten percent.
- Review wording of line 162.
- It is recommended to increase the font size of the graphics.
- In figure 1 indicate the definition of abbreviations. Final BW; (D) ADG; (E) ADFI; (F) FCR. Figures should be self-explanatory.
- Improve the quality and resolution of histology images in Figure 5.
- Indicate in the legend to Figure 6 the genes determined in each graph.
- In Figure 6 it is suggested to show only the graphs that had significant differences, to clean up the figure because it is congested. The values of the results that do not show differences should be included in some way in the text of item 3.5.
Author Response
- It is necessary to add quantifiable data on the effect of bile acids on lipid metabolism in the summary. The abstract is written in a very general way.
Thanks for your suggestion. We had indicated the cholesterol was lower, and triglycerides were higher in the HS birds (Line 43-44).
- In the summary indicate what the abbreviations are and then use them. AA, HS, TBA, ADFI, ADG, BW, and FCR???.
Thank you. All abbreviations have been defined in the abstract. Please see the re-submitted manuscript in red.
- Indicate how respiratory rate and cloacal temperature were measured.
As described by Zhen et al.,2015, respiratory rate was measured by counting the cloacal movements in a minute. Cloacal temperature was taken by a thermometer into the rectum. Reference has been added (Line 559-561).
- Line 140: change 10% by ten percent.
Changed as suggested.
- Review wording of line 162.
It has now been changed to: In the model, effects of temperatures (the main plot) and diets (the subplot) were fixed effects, while block and block × temperature were considered as random effects.
- It is recommended to increase the font size of the graphics.
Thanks for your suggestion. I have adjusted all the graphics.
- In figure 1 indicate the definition of abbreviations. Final BW; (D) ADG; (E) ADFI; (F) FCR. Figures should be self-explanatory.
All abbreviations have been defined in the figures.
- Improve the quality and resolution of histology images in Figure 5.
Thank you.
- Indicate in the legend to Figure 6 the genes determined in each graph.
Gene names have been added in each figure.
- In Figure 6 it is suggested to show only the graphs that had significant differences, to clean up the figure because it is congested. The values of the results that do not show differences should be included in some way in the text of item 3.5.
We have moved the Fig6. A, D, E, F, and G to supplementary materials.
- Add references for the following methods: thiobarbituric acid method, xanthine oxidase method.
References have been added. Reference has been added (Line 568-569).
- Add in the introduction references to the antioxidant capacity and antioxidant effect of bile acids.
Reviewer 4 Report
Major revision of the manuscript is needed as following:
Linguistic revision: We notice incorrect verb forms, wordy sentences, wrong choice of terms, etc.
Introduction: This has to be improved with the addition of more and recent references on poultry heat stress and nutritional strategies to minimize this.
Results: Probabilities must be added, even for non statistical differences observed.
Discussion: Transfer text to Introduction Section i.e. Lines 244-258, Lines 275-279, etc. Try to explain better the results obtained by this trial and why they are not as expected for all parameters under study.
Conclusion: Delete Lines 322-324 and improve your conclusions with the main output from this study.
Author Response
-Major revision of the manuscript is needed as following:
Linguistic revision: We notice incorrect verb forms, wordy sentences, wrong choice of terms, etc.
We have checked the language thoroughly. Thank you.
Results: Probabilities must be added, even for non statistical differences observed.
P values have been added.
Introduction: This has to be improved with the addition of more and recent references on poultry heat stress and nutritional strategies to minimize this.
References have been added. Please see the text (Line 81-89).
Discussion: Transfer text to Introduction Section i.e. Lines 244-258, Lines 275-279, etc. Try to explain better the results obtained by this trial and why they are not as expected for all parameters under study.
Thanks for your suggestion. Some parts of discussion have been re-organized to the introduction. Under thermoneutral environment, hepatic expression of genes related to lipid metabolism were not affected by BA supplements for 21 d. Neither were serum lipid, abdominal fat and growth performance. By compared with the two previous study, we think the time length for BA supplements was critical because only significant effects were observed until 42 d. However, BA supplement did lower liver fat and fat-related genes in the heat-stressed chickens after 21-d treatment. The finding suggested BA was more effective when animals have abnormal lipid metabolism.
Conclusion: Delete Lines 322-324 and improve your conclusions with the main output from this study.
Modified as suggested. Thank you.